# Iterative Learning-Based Negative Effect Compensation Control of Disturbance to Improve the Disturbance Isolation of System

**DOI:** 10.3390/s22093464

**Published:** 2022-05-02

**Authors:** Xiantao Li, Lu Wang, Xianqi Xia, Yuzhang Liu, Bao Zhang

**Affiliations:** Changchun Institute of Optics, Fine Mechanics and Physics, Chinese Academy of Sciences, Changchun 130033, China; wangluciomp@163.com (L.W.); xiaxianqi16@mails.ucas.edu.cn (X.X.); liuyuzhang1988@126.com (Y.L.); zhangb@ciomp.ac.cn (B.Z.)

**Keywords:** negative effect compensation of disturbance, off-line iterative learning control, stability control

## Abstract

At present, the cogging torque of permanent magnet synchronous motors (PMSM) seriously limits the Los pointing accuracy of aviation photoelectric stabilization platforms based on PMSM, which also restricts the requirements of ultra-long-distance and high-precision aviation reconnaissance and detection. For this problem, an off-line iterative learning control (ILC) was designed, and on this basis, a control method of negative effect compensation of disturbance (NECOD) is proposed. Firstly, the “dominant disturbance torque” in the system, that is, the cogging torque with the characteristics of position periodicity, was suppressed by off-line ILC according to different positions. Then, for the “residual disturbance” after compensation, NECOD was used to suppress it. In the constant speed scanning experiment of the aviation photoelectric stabilization platform, the method of combining the off-line iterative learning controller and the negative effect compensation of disturbance (NECOD + ILC) proposed in this paper significantly improved the Los control accuracy of the platform when compared with the classical active disturbance rejection control (ADRC) and ADRC + ILC methods, and the Los pointing error of the constant speed scanning process had only increased by less than 5% when the system had ±15% parameter perturbation. In addition, NECOD + ILC has fewer parameters and is easy to adjust, which is conducive to engineering application and promotion.

## 1. Introduction

Photoelectric stabilization platforms are widely used in visible, infrared and laser tracking systems in aerospace and aviation fields. It can effectively isolate the angular motion interference of carriers (missiles, aircraft, combat vehicles and ships), make the aiming device have high pointing accuracy and rapid mobility, reduce the image blur caused by flutter and improve the imaging quality [1,2,3]. However, with the increasing demand for long-range reconnaissance, the aperture, volume and weight of optical load are also increasing, as shown in Figure 1. Therefore, in order to improve the rapidity and stability of the photoelectric stability platform, the shafting radius of the system needs to be increased, and PMSM with higher torque output efficiency needs to be used to drive in order to achieve sufficient torque reserve [4,5], as shown in Figure 2. PMSM has high torque inertia ratio, high power density, high reliability and wide speed regulation range, so it is widely used in the fields of robotics, high-precision digital control machine tools, photoelectric turntables and so on [6].

PMSM mainly adopts vector control and direct torque control. Vector control can make the motor output torque proportional to the q-axis current by controlling the motor d-axis current to zero. Vector control with $i_{d}=0$ is the most widely used method of PMSM. Its algorithm is simple and has good torque characteristics. It is very suitable for low-speed working conditions. Therefore, this paper adopts the vector control scheme with $i_{d}=0$.

When compared with the traditional driving mode of DC torque motor, the influence of cogging torque on the visual axis stability of an optical lens is more obvious, and the cogging torque changes with the position of the motor relative to the stator. It can be seen that how to improve the suppression ability of position periodic disturbance is the key to improve the imaging quality of the system [7,8].

In the history of the development of disturbance suppression theory, it can be divided into “active disturbance rejection technology” and “passive disturbance rejection technology”. The disturbance suppression ability of “passive anti-interference technology” is directly related to the open-loop gain of the system in the disturbance frequency band, so the frequency range of disturbance suppression is limited by the open-loop crossing frequency of the system. The “active disturbance rejection technology” focuses on the active compensation of disturbance, which is more targeted and independent of the performance of the controlled system. When compared with “passive anti-interference technology”, its disturbance suppression ability is superior. During the development of “active anti-interference technology”, the following three methods have appeared:Absolute invariance principle

In the 1940s, Soviet scholars put forward the principle of absolute invariance “(in order to overcome the influence of external disturbance, the external disturbance must be measured, and the controller must contain both feedback stable channels and channels to suppress external disturbance, i.e., “dual channel principle”)”. The “dual channel principle” needs to measure the disturbance, which is difficult to realize in many systems. The disturbances such as cogging torque, friction torque and system parameter perturbation studied in this paper have many influencing factors and are coupled with each other. In the actual system, these disturbances are almost impossible be measured.

Internal model principle

In the 1970s, Canadian scholars put forward the “internal model principle” (if you want to overcome the influence of external disturbance, you must know the external disturbance model, and the controller must contain this external disturbance model). The disturbance torque is modeled based on the “internal model principle”, and the controller uses the disturbance model to generate control variables to suppress the disturbance. This method has been widely used [9,10,11,12].

Disturbance observer principle

The most prominent feature of the disturbance observer is that all the uncertain factors acting on the controlled object are reduced to “total unknown disturbance”, which is estimated and compensated for by using the input and output information of the system. There is no need to directly measure the external disturbance and predict the action law of the disturbance. In essence, it completely breaks through the limitations of “absolute invariance principle” and “internal model principle”. The representative examples are Disturbance Observer (DOB) and Active Disturbance Rejection Control (ADRC). However, the observation of disturbance by DOB [13,14,15,16] or ADRC [17,18,19,20] is essentially the reverse solution of the disturbance value, according to the result information caused by the disturbance. The necessary condition is that the disturbance has affected the system. Therefore, the improvement of disturbance suppression bandwidth is limited in principle. In addition, the parameter adjustment of ADRC has always been a thorny problem. Moreover, when the order of the system is greater than four, the speed of disturbance estimation is very slow, and it is difficult to give full play to the maximum potential of the controller.

Disturbance compensation can be divided into feedforward compensation, based on disturbance characteristics, and feedback compensation, based on real-time disturbance observation. The feedforward compensation method has high real-time performance, but it is easy to have overcompensation and under-compensation. The real-time observation compensation of disturbance has time lag, which leads to inaccurate compensation, especially in the medium and high frequency bands. In order to give full play to the advantages of the two compensation methods, ILC off-line compensation is designed to suppress the cogging torque of PMSM. In order to suppress the residual disturbance after ILC compensation and other real-time disturbances of the system, negative effect compensation control of disturbance is designed to estimate and compensate these total disturbances.

The iterative learning controller was first proposed by the Japanese scholar Uchiyama in 1978. It uses the past information of the system to design new control variables, and constantly corrects the unsatisfactory control variables by the deviation between the system output and the reference input, so as to obtain new and more ideal control variables, so as to finally realize the complete tracking of the system. ILC is used in various control systems because of its excellent ability to suppress periodic disturbances. The typical iterative learning controller is closed-loop control. According to the characteristics of ILC, the corresponding torque of the motor is obtained. During the actual operation of the system, the ILC is removed, and the obtained control variables are compensated according to the real-time position of the motor, so as to suppress the cogging torque. This method is referred to as “offline ILC” in this paper.

In this paper, ADRC and ADRC + ILC are used as the comparative control scheme, the aviation photoelectric stabilization platform driven by PMSM is used as the experimental equipment and the control accuracy of constant speed scanning is used as the comparative performance index. The experimental results show that the proposed scheme has stronger disturbance suppression ability and strong robustness to the perturbation of model parameters.

This paper is organized as follows. In Section 2, the mathematical model of the PMSM is described. Section 3 gives the specific design method and experimental results of ILC. Section 4 describes the specific design and analysis of NECOD, and designs a comparative experiment. Section 5 is the experimental results and analysis of stability and robustness experiment. The paper is concluded in Section 6.

## 2. Mathematical Model of PMSM

We considered a surface-mounted PMSM. Suppose that: (1) The magnetic flux of the motor is not saturated; (2) the eddy currents and hysteresis losses are negligible; (3) the three-phase stator windings of PMSM are sinusoidally distributed in space. For the purpose of control design, $\left(i_{d},\;i_{q},\;\omega \right)$ were chosen as state variables. The mathematical model of PMSM can be expressed as follow [21]:(1)$$\left(\begin{array}{c} {\dot{i}}_{d}\\ {\dot{i}}_{q}\\ \dot{\omega } \end{array}\right)=\left(\begin{array}{ccc} -R/L & p\omega & 0\\ -p\omega & -R/L & p\psi_{f}/L\\ 0 & K_{t}/J & -B/J \end{array}\right)\left(\begin{array}{c} i_{d}\\ i_{q}\\ \omega \end{array}\right)+\left(\begin{array}{c} u_{d}/L\\ u_{q}/L\\ -T_{d}/J \end{array}\right)\;$$
where, $i_{d},\;i_{q}$ are stator currents; $u_{d},\;u_{q}$ are stator voltages; R is resistance; L is inductance; $\psi_{f}$ is rotor flux; $K_{t}$ is torque coefficient; ω is mechanical angular speed; p is the pole pairs of motor; B is frictional coefficient; J is inertia; $T_{d}$ is the total disturbance torque borne by the system, which includes not only the cogging torque of the motor, but also the friction torque, mass unbalance torque and line disturbance torque of the system. FOC is the most commonly used control method in permanent magnet synchronous motors, where, the d-axis current is controlled to 0, and the output torque of the motor is directly proportional to the q-axis current.

## 3. Design of Iterative Learning Controller

### 3.1. Use ILC to Specifically Suppress Cogging Torque in the Motor

Because the cogging torque in the motor is a periodic disturbance of position, ILC was adopted in the spatial domain in this paper. That is, the final torque output of the controller is obtained by using the last control variable at the same position in the motor and the current system error. The last control variable in the same position is used to record the periodic control information of motor position, while the current error of the system contains aperiodic information. Thus, ILC uses the recorded position periodic information to suppress periodic interference. The principle block diagram is shown in Figure 3 [22,23]:

The ILC control law adopted is as follows:(2)$$u_{k}(\theta )=u_{k-1}(\theta )+G_{P}e_{k}(\theta )+G_{I}\int e_{k}(\theta )dt$$
where, the subscript k is the number of iterations; θ is the current mechanical angle of the motor; $\omega_{d}$ is the desired angular velocity of the system; ω_*t*_ is the angular velocity of the actual movement of the system, which is directly measured by the gyroscope; u is the output of ILC controller; d is the disturbance torque. According to Figure 3, the system output is:(3)$$\omega_{k}=Pu_{k}+d_{k}$$
where P is the system transfer function. Substitute Formula (2) into Formula (3), and d is a periodic function of position because the cogging torque in the motor, that is $d_{k}(\theta )=d_{k-1}(\theta )$, the tracking error of the *k*th iteration is as follows:(4)$$\begin{array}{ll} e_{k} & =e_{k-1}-P\left(G_{P}e_{k}+G_{I}\int e_{k}dt\right)+d_{k-1}-d_{k}\\ & =e_{k-1}-P\left(G_{P}e_{k}+G_{I}\int e_{k}dt\right) \end{array}$$

It can be seen that the tracking error of the system is independent of the disturbance, and the external disturbance is completely suppressed.

Make the system rotate at a uniform angular speed at the set speed in the closed-loop state, and adjust the gain of the controller to make the running speed stable. At this time, the motor is in the torque balance state, that is, the output torque of the motor is equal to the sum of the system resistance. At this time, this is mainly used to overcome the friction torque and cogging torque. We set the clockwise rotation direction as positive and the counterclockwise rotation direction as negative. When the motor rotates clockwise and counterclockwise at a uniform speed, the expression is as follows:(5)$$Ku_{\pm }(\theta )=T_{c}(\theta )\pm T_{f}$$
where, $u_{+}(\theta )$ and $u_{-}(\theta )$ are the stator voltages when the system is in the position θ, *K* is the ratio coefficient between the stator voltages and the output torque of the motor, $T_{c}(\theta )$ is the cogging torque at the angle θ and $T_{f}$ is the shaft friction torque. From Equation (5), $T_{c}(\theta )$ is easy to obtain as follows:(6)$$T_{c}(\theta )=\frac{K\left(u_{+}(\theta )+u_{-}(\theta )\right)}{2}$$

Obviously, the average value of the corresponding control variable is the cogging torque when the system rotates at a constant speed clockwise and counterclockwise. Feed forward the control variable corresponding to the cogging torque to the system, which can eliminate the influence of the motor cogging torque on the control accuracy of the system. Of course, because there are other forms of interference in the actual system, and it is impossible for the motor to maintain an absolute uniform speed during rotation, this method has some errors. For these deviations, the methods described later can be used to suppress them.

### 3.2. Specific Design for ILC

The desired angular speed of the platform rotation was set to 10°/s. During the rotation, the corresponding control variables in the ILC were collected every 1°. No matter how the motor rotates, to any position, the current control variable can be obtained through the linear interpolation of the control variable in the last iteration:(7)$$u_{k-1}(\theta )=\left(u_{k-1}\left([\theta ]+1\right)-u_{k-1}\left(\left[\theta \right]\right)\right)\cdot \left(\theta -[\theta ]\right)+u_{k-1}\left(\left[\theta \right]\right)$$
where, [θ] is the value that the angle value is rounded down, and $u_{k-1}\left(\left[\theta \right]\right)$ represents the corresponding control variable at the angle θ downward rounding of the (k−1)th iteration.

In the actual system the repeatability of the interference torque in the spatial domain is not absolute, that is, $d_{k}$ and $d_{k+1}$ are only approximately equal. In this case, the system will have errors, which will gradually accumulate and eventually lead to system instability. To solve this problem, the forgetting factor α is added on the basis of Equation (2):(8)$$u_{k}(\theta )=\left(1-\alpha \right)u_{k-1}(\theta )+G_{P}e_{k}(\theta )+G_{I}\int e_{k}(\theta )dt$$

At this time, the iterative error of kth is:(9)$$e_{k}=\left(1-\alpha \right)e_{k-1}-F\left(e_{k}\right)+\alpha \omega_{d}+\left(1-\alpha \right)d_{k-1}-d_{k}$$

Obviously, the introduction of forgetting factor can reduce the cumulative error caused by aperiodic interference. The improved system block diagram is as Figure 4:

### 3.3. ILC Experiment

Based on the above ideas, the iterative learning controller of the system was designed and tested. In the aviation photoelectric stabilization platform of this experiment, the execution frequency of current loop was 8 kHz and that of speed loop was 1 kHz. In order to test the effectiveness of ILC in the experimental system, ILC was used at the mechanical angle 0~200°, and the PI controller was used at the rest positions. The desired angular speed of the platform rotation was set to 10°/s. The value of α was set to 0.07. Experimental results are as shown in Figure 5:

It is obvious from the experimental results that ILC can reach the speed error of ±0.5°/s after the third iteration. It can be seen that for the position periodic disturbance, the suppression effect of ILC is obvious, due to the PI controller.

### 3.4. Control Variable Feedforward to Restrain Cogging Effect

The cogging torque is determined by the motor process, and its size is hardly affected by the working environment of the system. According to the method proposed in Section 3.1 and Section 3.2, we obtained the variation curve of motor cogging torque with mechanical angle, as shown in Figure 6.

Where $u_{cog}$ is an array of size 360. Read the mechanical angle value θ in real time during the operation of the motor, find the adjacent control variables $u_{cog}\left(\left[\theta \right]\right)$ and $u_{cog}\left(\left[\theta +1\right]\right)$ from the $u_{cog}$ array, and calculate the control variable corresponding to the cogging torque at angle t according to Equation (7). By compensating this control variable into the control loop, the cogging torque can be restrained, as shown in Figure 7.

## 4. Design and Analysis of Negative Effect Compensation Control of Disturbance

### 4.1. Control Strategy of Negative Effect Compensation of Disturbance

Based on the ILC compensation mentioned above, it is inevitable that there will be “under-compensation” and “overcompensation”. A control method for compensating the negative effect of disturbance is proposed to suppress the “residual sum disturbance” after ILC compensation. This method does not depend on any information of the system model, and the parameter adjustment is simple, which is very easy to be applied in reengineering. The specific principle is as follows.

In the ideal case (ignoring the disturbance), the closed-loop working principle block diagram of the system is shown in the Figure 8:

Where *G*(*s*) is the controller model, *P*(*s*) is the system controlled model, *R*(*s*) is the expected input of the system and *C*(*s*) is the actual output of the system. The transfer function of the system is:(10)$$C(s)=\frac{G(s)P(s)}{1+G(s)P(s)}R(s)$$

The error transfer function of the system is:(11)$$E(s)=\frac{1}{1+G(s)P(s)}R(s)$$

However, under the influence of disturbance torque, the output of the system will deviate from the input, and its action principle block diagram is shown in the Figure 9:

Where *D*(*s*) is the “sum disturbance” borne by the system. At this time, the transfer function of the system is:(12)$$C(s)=\frac{G(s)P(s)}{1+G(s)P(s)}R(s)+\frac{1}{1+G(s)P(s)}D(s)P(s)$$

The error transfer function of the system is:(13)$$E(s)=\frac{1}{1+G(s)P(s)}R(s)+\frac{1}{1+G(s)P(s)}D(s)P(s)$$

When compared with the ideal situation, the first term is the steady-state error term of the system, and the second term is the error term caused by torque interference. The purpose of disturbance suppression is to reduce the influence of the second term as much as possible. When combined with the above, the “dominant disturbance torque” of the system is firstly restrained under the compensation of ILC, and its “residual disturbance” is equivalent to the disturbance *D*(*s*) stated in this paper. This paper aimed to find a control method to eliminate the influence of “residual disturbance” *D*(*s*) on the system without introducing additional measurements.

By integrating and simplifying the system block diagram shown in the Figure 9, we can obtain the diagram as Figure 10.

In order to eliminate the influence of interference on the system without changing the system controller *g*(*s*), two additional feedback links were introduced. The principle block diagram is shown in the Figure 11:

Where 1/z is the delay module, the delay time is set to τ, and the transfer function of the delay link is:(14)$$G_{1}(s)=\frac{1}{1-e^{-\tau s}}$$

The transfer function of the system at this time is deduced:(15)$$C(s)=\frac{\frac{1}{1-e^{-\tau s}}\frac{G(s)P(s)}{1+G(s)P(s)}}{1+\frac{1}{1-e^{-\tau s}}\frac{G(s)P(s)}{1+G(s)P(s)}}R(s)$$
(16)$$C(s)=\frac{G(s)P(s)}{\left(1-e^{-\tau s})+(2-e^{-\tau s})G(s)P(s\right)}R(s)$$

When the expected input of the system is zero and only the influence of disturbance term is considered:(17)$$C(s)={((e}^{-\tau s}-2)\frac{G(s)P(s)}{\left(1-e^{-\tau s})+(2-e^{-\tau s})G(s)P(s\right)}+1)D(s)P(s)$$

In this case, the transfer function of the system is:(18)$$C(s)=\frac{G(s)P(s)}{\left(1-e^{-\tau s})+(2-e^{-\tau s})G(s)P(s\right)}R(s){+((e}^{-\tau s}-2)\frac{G(s)P(s)}{\left(1-e^{-\tau s})+(2-e^{-\tau s})G(s)P(s\right)}+1)D(s)P(s)$$

The error transfer function of the system is:(19)$$E(s)=\frac{\left(1-e^{-\tau s})(G(s)P(s)+1\right)}{\left(1-e^{-\tau s})+(2-e^{-\tau s})G(s)P(s\right)}R(s){+((e}^{-\tau s}-2)\frac{G(s)P(s)}{\left(1-e^{-\tau s})+(2-e^{-\tau s})G(s)P(s\right)}+1)D(s)P(s)$$

When the delay time τ is small (the control period in general digital system is less than 1 ms) [24], according to Taylor expansion, we ignore the higher-order term. At this time, the delay link can be approximately:(20)$$e^{-\tau s}=1-\tau s$$

Thus, the above formula can be simplified as:(21)$$C(s)=\frac{G(s)P(s)}{\tau s+(1+\tau s)G(s)P(s)}R(s)+((-1-\tau s)\frac{G(s)P(s)}{\tau s+(1+\tau s)G(s)P(s)}+1)D(s)P(s)$$
(22)$$C(s)=\frac{G(s)P(s)}{\tau s+(1+\tau s)G(s)P(s)}R(s)+\frac{\tau s}{\tau s+(1+\tau s)G(s)P(s)}D(s)P(s)$$
(23)$$E(s)=\frac{-\tau s(G(s)P(s)+1)}{\tau s+(1+\tau s)G(s)P(s)}R(s)+\frac{\tau s}{\tau s+(1+\tau s)G(s)P(s)}D(s)P(s)$$

Obviously, the smaller the delay time τ, the stronger the disturbance suppression ability. If the delay time approaches 0, then
(24)E(s)=0

It can be found that at this time, the influence of the disturbance term is eliminated, and the system principle block diagram can be simplified as Figure 12:

On this basis, considering the influence on the system in the presence of other interference, d(*s*) was introduced as the measurement error, and its principle block diagram is shown in the Figure 13:

When only d(*s*) is considered, its impact on the system is discussed:(25)$$C(s)={(e}^{-\tau s}-2)\frac{G(s)P(s)}{\left(1-e^{-\tau s})+(2-e^{-\tau s})G(s)P(s\right)}d(s)$$

If the delay time τ approaches 0, $e^{-\tau s}$ will approach 1 and substitute into the above formula:(26)C(s)=−d(s)

It can be seen that although the measurement of the system has an impact on the output of the system, there is no differential link in NECOD, so the impact of measurement error on the output of the system has not been amplified.

Now we discuss the influence of model parameter perturbation on the system. Assuming that there is modeling error, the error percentage is δ, and the system principle block diagram is shown in the Figure 14:

Here, the transfer function of the system is:(27)$$C(s)=\frac{\left(1+\delta )G(s)P(s\right)}{\left(1-e^{-\tau s})+(2-e^{-\tau s})(1+\delta )G(s)P(s\right)}R(s){+((e}^{-\tau s}-2)\frac{{\left(1+\delta \right)}^{2}G(s)P(s)}{\left(1-e^{-\tau s})+(2-e^{-\tau s})(1+\delta )G(s)P(s\right)}+(1+\delta ))D(s)P(s)$$

The error transfer function of the system is:(28)$$E(s)=\frac{-(1-e^{-\tau s})(1+\delta )G(s)P(s)-(1-e^{-\tau s})}{\left(1-e^{-\tau s})+(2-e^{-\tau s})(1+\delta )G(s)P(s\right)}R(s){+((e}^{-\tau s}-2)\frac{{\left(1+\delta \right)}^{2}G(s)P(s)}{\left(1-e^{-\tau s})+(2-e^{-\tau s})(1+\delta )G(s)P(s\right)}+(1+\delta ))D(s)P(s)$$
(29)$$E(s)=\frac{-\tau s((1+\delta )G(s)P(s)+1)}{\tau s+(1+\tau s)(1+\delta )G(s)P(s)}R(s)+\frac{(1+\delta )\tau s}{\tau s+(1+\delta )(1+\tau s)G(s)P(s)}D(s)P(s)$$

If the delay time τ approaches 0, $e^{-\tau s}$ will approach 1 and we substitute into the above formula:(30)E(s)=0

The change of model parameters has little effect on the system transfer function and anti-interference performance.

In order to further adjust in combination with the characteristics of the controlled object during commissioning, a coefficient B was added to the feedback loop, as shown in Figure 15:

Here:(31)$$C(s)=\frac{G(s)P(s)}{\left(1-be^{-\tau s})+(1+b-be^{-\tau s})G(s)P(s\right)}R(s)+((be^{-\tau s}-2)\frac{G(s)P(s)}{\left(1-be^{-\tau s})+(1+b-be^{-\tau s})G(s)P(s\right)}+1)D(s)P(s)$$

Similarly, according to the above principle, if the delay time τ approaches 0, then:(32)$$C(s)=\frac{G(s)P(s)}{\left(1-b+b\tau s))+(1+b\tau s)G(s)P(s\right)}R(s)+\frac{\left(1-b+b\tau s)+(b-1)G(s)P(s\right)}{\left(1-b+b\tau s)+(1+b\tau s)G(s)P(s\right)}D(s)P(s)$$

Obviously, when *b* is closer to 1, the disturbance suppression ability of the system is stronger. Therefore, the value of *b* can be further optimized in combination with the stability of the system during commissioning.

As a comparison, two representative control algorithms based on disturbance observation principle, ADRC and DOB, are described below. The specific principle block diagram is shown in the Figure 16.

### 4.2. Control Scheme for Comparison

Among them, taking ADRC as an example for principle analysis. ADRC is a new control technology, and its core idea is “active disturbance rejection”. It collectively refers to the influence of the external disturbance of the system and the uncertainty of the system model on the system as “unknown disturbance”, and then estimates the “unknown disturbance” in real time through the extended state observer (ESO), so as to realize the direct feedforward compensation control of the disturbance. Then, the purpose of improving disturbance isolation is achieved [25,26].

Combining the control object parameters obtained by current loop processing and system identification, Equation (1) is rewritten into the form of state equation:(33)$$\left\{\begin{array}{l} {\dot{x}}_{1}=31u+d\\ y=x_{1} \end{array}\right.$$
where, *d* represents the sum disturbance in Figure 9, and *d* in Equation (34) is added to the system as a new, extended state, that is:(34)$$x_{2}=d$$

And let:(35)$${\dot{x}}_{2}=a(t)$$
where *a*(*t*) is the change rate of “unknown disturbance”, so Equation (34) can be rewritten as:(36)$$\left\{\begin{array}{l} {\dot{x}}_{1}=x_{2}+31u\\ {\dot{x}}_{2}=a(t)\\ y=x_{1} \end{array}\right.$$

The first-order controlled object adds another extended state, that is, system disturbance d. Thus, the total disturbance d can be estimated by designing a second-order ESO for the expanded object. In this paper, the second-order nonlinear extended state observer was used to observe the disturbance d to realize $z_{2}\to d$:(37)$$\left\{\begin{array}{l} e_{1}=z_{1}-y\\ {\dot{z}}_{1}=z_{2}-\beta_{01}e_{1}+bu\\ {\dot{z}}_{2}=-\beta_{02}fal(e_{1},\frac{1}{2},\delta )\\ y=z_{1} \end{array}\right.$$
where:(38)fal(e,α,δ)=eδα−1e≤δesign(e)e>δ

Here, whether the sum disturbance *d* is continuous or discontinuous, known or unknown; as long as it is bounded in the process, we can always choose the appropriate parameters $\beta_{01},\;\beta_{02}$, so that the extended state observer can estimate the state $x_{1}$ and the extended state $x_{2}$ of the system in real time. In setting parameters $\beta_{01},\;\beta_{02}$, Dr. Gao Zhiqiang of Cleveland State University gave a simple method to determine the parameters of linear extended state observer by using the concept of bandwidth [27]. For the first-order controlled object, it is configured as ${\left(s+\omega \right)}^{2}$. Accordingly, it can be concluded that $\beta_{01}=2\omega ,\beta_{02}=\omega^{2}$. Generally, the selection range of ω is very large, so it is easy to adjust the appropriate one ω.

Obviously, when compared with the classical ADRC, the NECOD method has the following advantages:It is completely independent of the system model and can be designed independently.No differential link, strong noise adaptability.Few parameters, simple adjustment and easy engineering practice.

### 4.3. ILC Combined with NECOD

The core idea of this paper for disturbance suppression in the system is: ILC was used to suppress the cogging torque of the motor. On this basis, the NECOD strategy was used to suppress the “residual disturbance” in the system. This compliance control strategy is called NECOD + ILC. The specific control block diagram is as Figure 17.

As a comparative experiment, ADRC, which is a representative of the “active anti-interference” strategy, and the scheme of ADRC combined with offline ILC in this paper were adopted. The speed stability and Los pointing error of ADRC, ADRC + ILC and NECOD + ILC proposed in this paper were compared in constant speed scanning. These experiments were carried out in the same aviation photoelectric stabilization platform, as shown in Figure 18.

### 4.4. Controller Parameters

After parameter tuning, under the same controller *G*(*s*), the controller parameters of NECOD and ADRC were as follows: the control parameter of NECOD was 0.95. Parameter of ADRC was ω=300. 2δ was the interval length of the linear segment in the function, which can avoid the high-frequency oscillation of ESO near the origin. The platform used a fiber optic gyroscope to measure the angular speed, and its noise amplitude was 0.02°/s. The value of δ should be greater than the noise peak of the gyro, and 0.03 was selected in this experiment.

## 5. Experimental Results and Discussion

The experimental condition was trapezoidal wave tracking of the swing table, and the scanning speed fluctuation of the system under three control strategies was compared. The ascending acceleration of trapezoidal wave was $100^{\circ }/s^{2}$, the uniform scanning speed was 60°/s and the constant speed holding time was 2 s.

### 5.1. Static Experiment

Trapezoidal wave tracking was carried out under the static condition of a flight simulation swing table. The experiment was repeated five times, and the difference between the results was small. The typical experimental result of one of them is shown in Figure 19:

It can be seen that the control strategy of NECOD + ILC proposed in this paper has less overshoot than ADRC + ILC and ADRC.

In the actual work, the platform needs to scan at a uniform speed of 1.5 s. The Los pointing error of the optical load in the scanning process determines the false alarm rate of the aviation photoelectric stabilization platform for the detection of weak and small targets. Take 1.1~2.6 s and 4.3~5.8 s of each trapezoidal wave tracking as the forward and reverse scanning process, respectively.

Integrating the speed error to obtain the Los pointing error:(39)$$\Delta \theta (t)=\int \left(\omega_{d}-\omega \right)dt$$

The sampling rate of the experimental data in this paper was 1 kHz. For discretization of the above formula, we can get:(40)$$\Delta \theta =\sum \left(\omega_{d}-\omega \right)\cdot 0.001$$

The variation curves of speed error and Los pointing error during forward and reverse scanning are as Figure 20:

The Los pointing error in the scanning process in the five experiments was statistically analyzed, and the results are shown in Table 1:

From the statistical results of Los pointing error, it can be seen that when compared with the classical ADRC, the Los pointing accuracy of the system has been significantly improved after the introduction of off-line ILC. When compared with the classical ADRC and ADRC + ILC, the average value and RMS value of Los pointing error of NECOD + ILC proposed in this paper have been significantly improved, which proves the effectiveness and superiority of NECOD + ILC in the control of aviation photoelectric stabilization platform.

### 5.2. Dynamic Experiment

We set the swing table to the amplitude of 2° and frequency of 2 Hz for swing, repeated the experiment outlined in the previous section and calculated the system Los pointing error based on three control schemes under swing conditions. The experimental results are shown in Figure 21 and Table 2.

It can be seen from the swing experiment that the system based on NECOD + ILC control was less affected by the swing disturbance, and its Los pointing error does not increase significantly when compared with the static condition. The suppression performance of NECOD + ILC proposed in this paper is proved. When compared with the classical ADRC, the NECOD + ILC proposed in this paper improves the Los stability accuracy of the system by at least one time. When compared with ADRC + ILC, the performance has also been improved, but on the whole, the difference between the two is not very great.

### 5.3. Controller Robustness Experiment

In order to test the robustness of NECOD, the parameters of the controlled object were changed by artificially increasing the load, so that the change of parameter *k* in Equation (37) was ±15%. Then, we repeated the uniform scanning experiment and calculated the RMS value of Los pointing error. The experimental results are shown in Table 3.

From the robustness experiment, it can be seen that when the system model has ± 15% parameter perturbation, the NECOD + ILC control scheme still maintained high control performance, and its Los stability error was only increased by less than 5% when compared with that without perturbation, which proves the robustness of NECOD + ILC control scheme to model change.

## 6. Conclusions

The off-line ILC compensation method proposed in this paper can effectively suppress the cogging torque of PMSM, and its implementation is simple and will not affect the stability of the system.

From the trapezoidal wave tracking experiment of the platform, it can be seen that the NECOD + ILC control strategy proposed in this paper has less overshoot than the classical ADRC. During the forward and reverse scanning at the speed of 60°/s, the RMS value of the Los pointing error of the system was less than 0.0046° within 1.5 s, which fully meets the actual working requirements of the system. When compared with the classical ADRC and ADRC + ILC, the Los pointing error had significantly reduced. In the robustness experiment, when the parameters were perturbed by ±15%, the Los pointing error increased by less than 5% in the process of uniform scanning, which proves that NECOD + ILC has strong robustness.

To summarize, the NECOD + ILC control scheme proposed in this paper has the advantages of simple parameter setting, easy engineering implementation and strong robustness. It is a very superior control scheme for PMSM driven aviation photoelectric stability platform. This control scheme has been verified by experiments, which fully meets the actual working requirements of the system, and also lays a foundation for the subsequent research of higher precision control scheme of the platform.

## Figures and Tables

**Figure 1 sensors-22-03464-f001:**
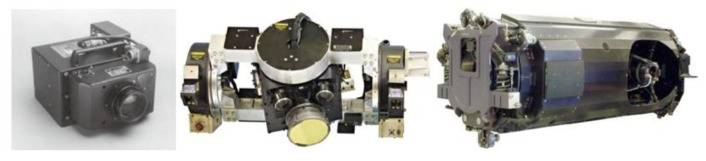
The volume and weight of optical load increase gradually.

**Figure 2 sensors-22-03464-f002:**
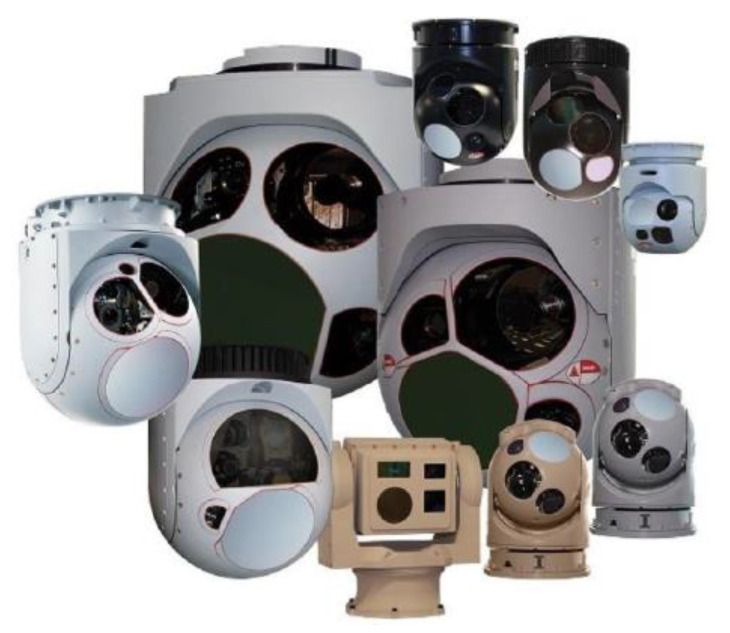
With the increase of platform volume, the shafting size also increases (30~180 mm).

**Figure 3 sensors-22-03464-f003:**
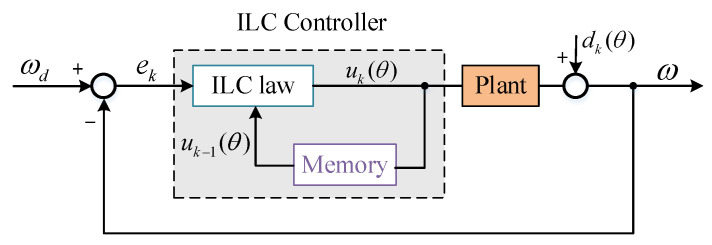
Principle block diagram of ILC.

**Figure 4 sensors-22-03464-f004:**
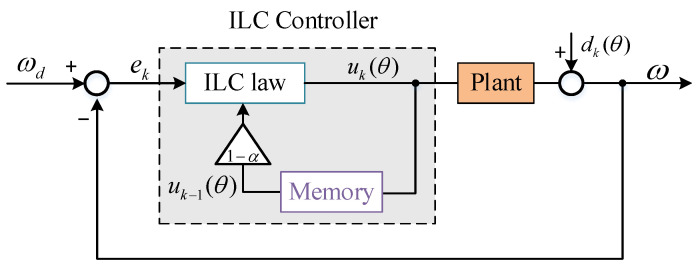
Block diagram of improved ILC.

**Figure 5 sensors-22-03464-f005:**
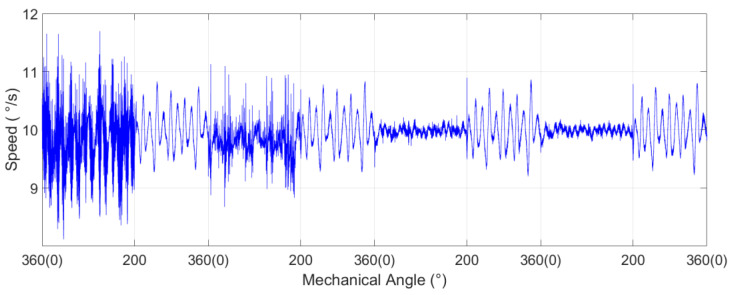
Experimental results of ILC.

**Figure 6 sensors-22-03464-f006:**
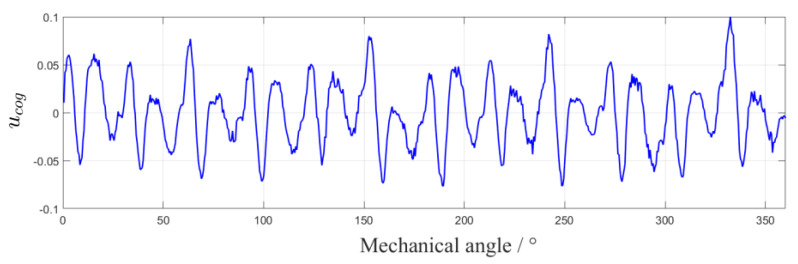
Variation curve of u with mechanical angle.

**Figure 7 sensors-22-03464-f007:**
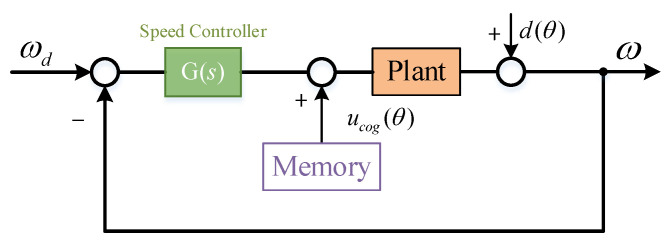
System diagram after ILC feedforward.

**Figure 8 sensors-22-03464-f008:**
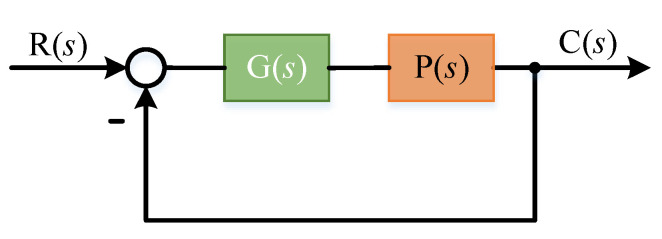
Typical closed-loop control block diagram.

**Figure 9 sensors-22-03464-f009:**
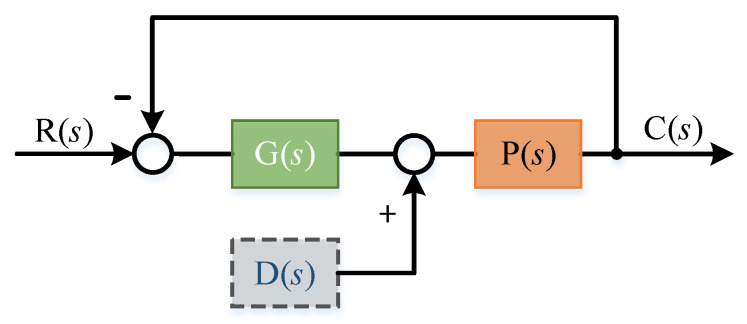
Closed loop control block diagram in the presence of disturbance.

**Figure 10 sensors-22-03464-f010:**
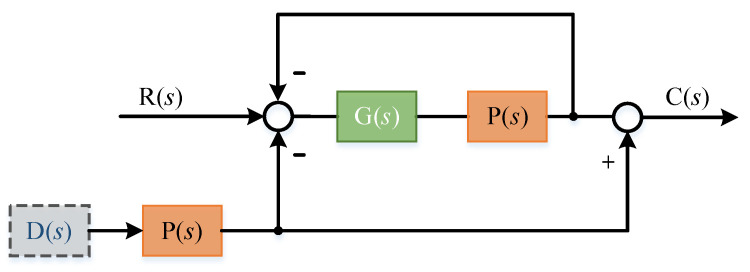
Equivalent block diagram in the presence of disturbance.

**Figure 11 sensors-22-03464-f011:**
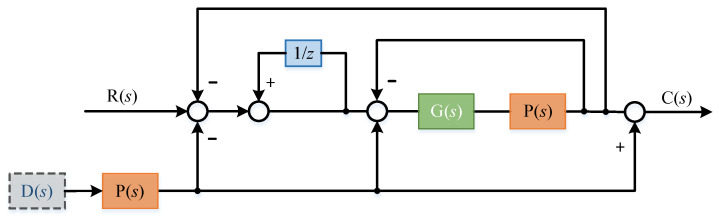
Two feedback links re introduced to eliminate external disturbance.

**Figure 12 sensors-22-03464-f012:**
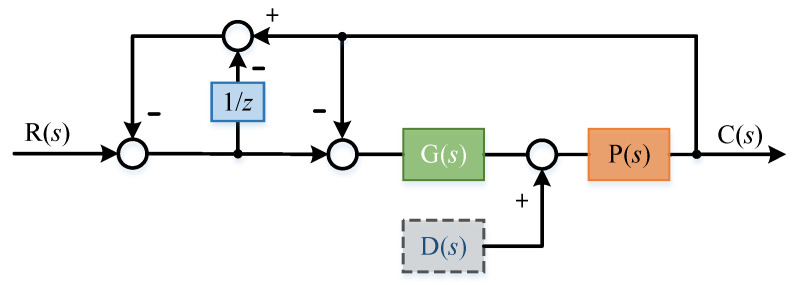
System block diagram when delay is 0.

**Figure 13 sensors-22-03464-f013:**
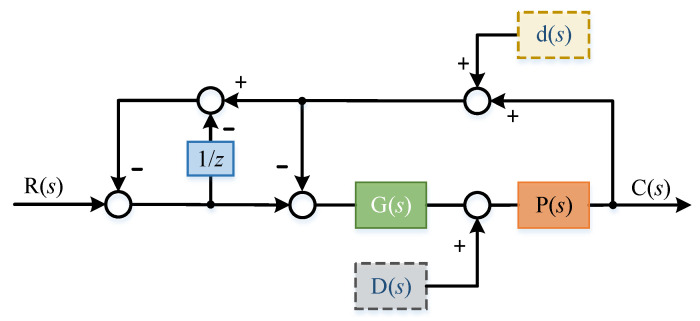
d(*s*) is introduced as the measurement error.

**Figure 14 sensors-22-03464-f014:**
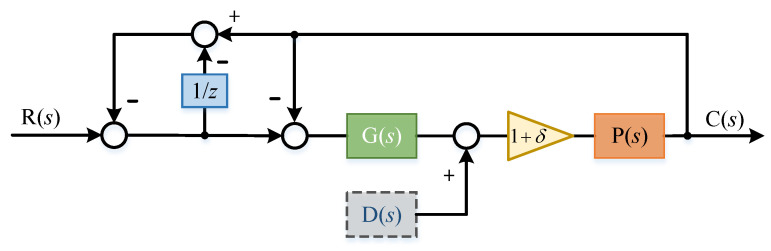
System block diagram with parameter perturbation.

**Figure 15 sensors-22-03464-f015:**
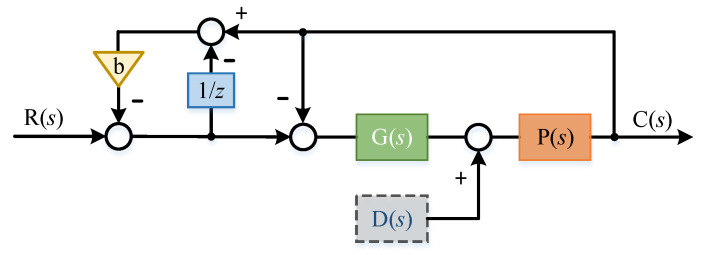
Parameter b is introduced for further adjustment.

**Figure 16 sensors-22-03464-f016:**
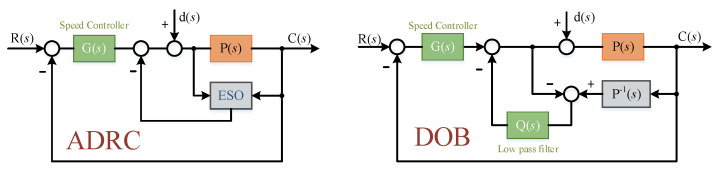
Principle block diagram of ADRC and DOB.

**Figure 17 sensors-22-03464-f017:**
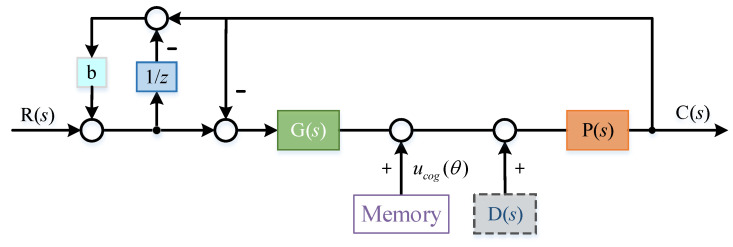
Principle block diagram of NECOD + ILC.

**Figure 18 sensors-22-03464-f018:**
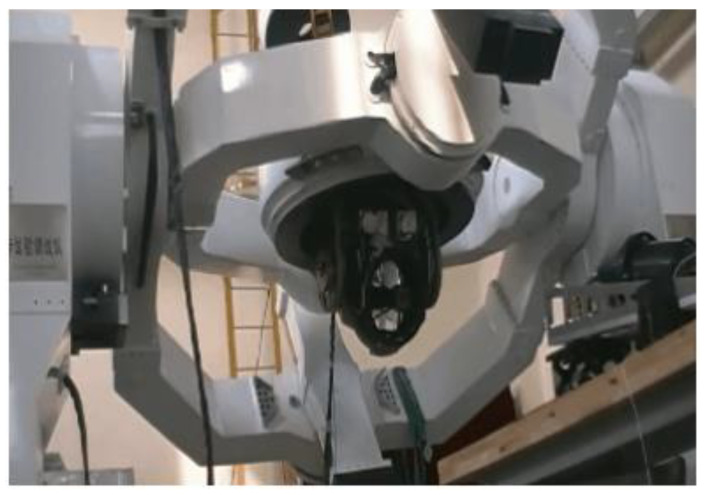
Swing table experiment of aviation photoelectric stabilization platform.

**Figure 19 sensors-22-03464-f019:**
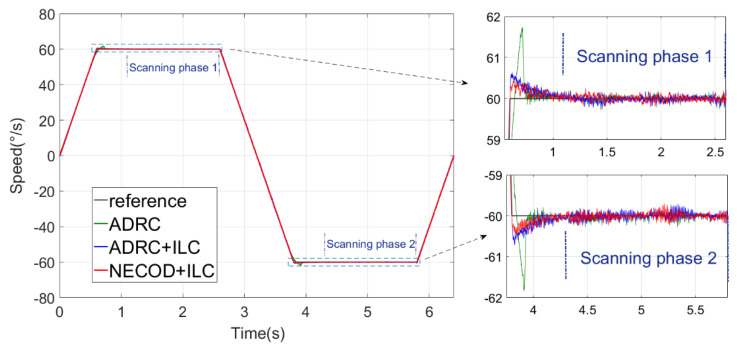
Experimental results of trapezoidal wave tracking under static conditions.

**Figure 20 sensors-22-03464-f020:**
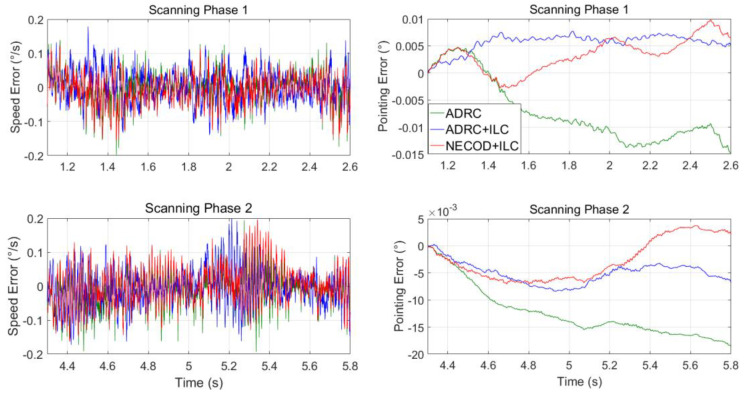
Speed fluctuation and Los pointing error under static conditions.

**Figure 21 sensors-22-03464-f021:**
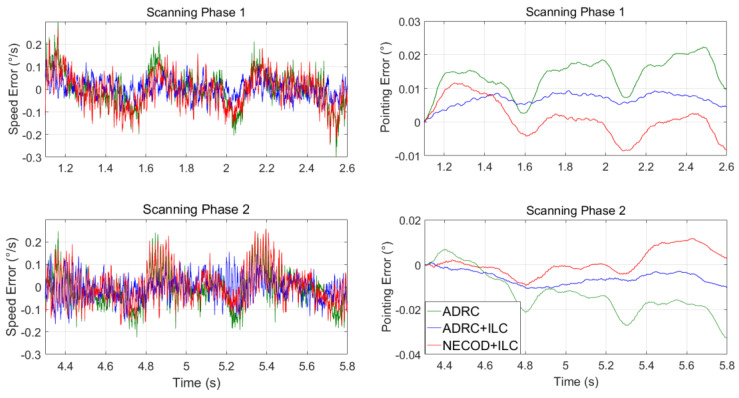
Speed fluctuation and Los pointing error under swing conditions.

**Table 1 sensors-22-03464-t001:** Comparison of Los pointing errors under static conditions.

**Scanning Phase 1**			
	$\max\left(\left|\Delta \theta \right|\right)$	$\mathrm{mean}\left(\left|\Delta \theta \right|\right)$	$\mathrm{rms}\left(\left|\Delta \theta \right|\right)$
ADRC	0.0149°	0.00837°	0.00925°
ADRC + ILC	0.00772°	0.00563°	0.00585°
NEODC + ILC	0.00983°	0.00383°	0.00449°
**Scanning Phase 2**			
	$\max\left(\left|\Delta \theta \right|\right)$	$\mathrm{mean}\left(\left|\Delta \theta \right|\right)$	$\mathrm{rms}\left(\left|\Delta \theta \right|\right)$
ADRC	0.0186°	0.0124°	0.0132°
ADRC + ILC	0.00837°	0.00500°	0.00538°
NEODC + ILC	0.00704°	0.00417°	0.00459°

**Table 2 sensors-22-03464-t002:** Comparison of Los pointing errors under swing conditions.

**Scanning Phase 1**			
	$\max\left(\left|\Delta \theta \right|\right)$	$\mathrm{mean}\left(\left|\Delta \theta \right|\right)$	$\mathrm{rms}\left(\left|\Delta \theta \right|\right)$
ADRC	0.0221°	0.0138°	0.0146°
ADRC + ILC	0.00934°	0.00670°	0.00694°
NEODC + ILC	0.0116°	0.00396°	0.00527°
**Scanning Phase 2**			
	$\max\left(\left|\Delta \theta \right|\right)$	$\mathrm{mean}\left(\left|\Delta \theta \right|\right)$	$\mathrm{rms}\left(\left|\Delta \theta \right|\right)$
ADRC	0.0329°	0.0137°	0.0156°
ADRC + ILC	0.0105°	0.00586°	0.00653°
NEODC + ILC	0.0116°	0.00400°	0.00541°

**Table 3 sensors-22-03464-t003:** Comparison of Los pointing error with parameter perturbation of ±15%.

**Scanning Phase 1**		
	$\Delta =15\%,\;\mathrm{rms}\left(\left|\Delta \theta \right|\right)$	$\Delta =-15\%,\;\mathrm{rms}\left(\left|\Delta \theta \right|\right)$
ADRC	0.0103°	0.00973°
ADRC + ILC	0.00641°	0.00650°
NEODC + ILC	0.00467°	0.00473°
**Scanning Phase 2**		
	$\Delta =15\%,\;\mathrm{rms}\left(\left|\Delta \theta \right|\right)$	$\Delta =-15\%,\;\mathrm{rms}\left(\left|\Delta \theta \right|\right)$
ADRC	0.0140°	0.0132°
ADRC + ILC	0.00587°	0.00537°
NEODC + ILC	0.00471°	0.00483°

## Data Availability

Not applicable.

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
