# Peer review of "Iterative Learning-Based Negative Effect Compensation Control of Disturbance to Improve the Disturbance Isolation of System"

_sensors, 2022, doi:10.3390/s22093464_

Round 1
Reviewer 1 Report
Thank you for giving me the opportunity to review the paper titled "Iterative Learning Based Negative Effect Compensation Control of Disturbance to Improve the Disturbance Isolation of System". The paper focuses on Iterative Learning control and negative effect compensation of disturbance. The following points should be improved in the paper:
- give a brief description of the paper organization at the end of the introduction section,
- There is no literature overview in the introduction part. Please refer to similar approaches related to ILC, disturbance compensation, and control structures of PMSM. In my opinion, one of the interesting comparisons should be pointed to: "State Feedback Speed Control with Periodic Disturbances Attenuation for PMSM Drive" (it was proved, that state feedback controller is better than DOBC+PI controller)
- Please prove a schematic diagram of the proposed control structure,
- Please provide a brief description that summarizes the required steps to design the proposed control structure. The ILC is tuned off-line, so in my opinion, it is necessary to point out the overall mechanism.
- in the text, you refer to some important assumptions or knowledge without citation, e.g., line 270-271, line 360-361
- To the best of my knowledge, the ILC is used to minimize the error to zero in the predefined window. In such a case, I do not understand what the ILC is used to, due to the fact that you obtained overshoot. ILC allows obtaining perfect tracking.
- What was the goal of the proposed control? Please improve the definition, of what you want to achieve by using the developed method. You mention about compensation of cogging torque in PMSM drive, But you do not present the comparison to clear PI control structure without compensation. I am asking for this because I do not understand the relation of torque load for checking the robustness.
- There is no additional plot in robustness experiments.
Author Response
We are truly grateful to the reviewers’ suggestions. Based on these comments, we have made careful modifications on the manuscript.
Point 1: give a brief description of the paper organization at the end of the introduction section
Response 1: We added a brief introduction to the structure of the paper at the end of section 1 of the revised manuscript.
Point 2: There is no literature overview in the introduction part. Please refer to similar approaches related to ILC, disturbance compensation, and control structures of PMSM. In my opinion, one of the interesting comparisons should be pointed to: "State Feedback Speed Control with Periodic Disturbances Attenuation for PMSM Drive" (it was proved, that state feedback controller is better than DOBC+PI controller)
Response 2: In the revised manuscript, we added the literature review of ILC, disturbance compensation and PMSM control in the introduction. Please refer to Line 46~51, Line 99~120 of the revised version for details
Point 3: Please prove a schematic diagram of the proposed control structure
Response 3: In Section 4.3, we give the block diagram of the control scheme proposed in this paper.
Point 4: Please provide a brief description that summarizes the required steps to design the proposed control structure. The ILC is tuned off-line, so in my opinion, it is necessary to point out the overall mechanism.
Response 4: In Section 3.4 of the revised manuscript, we describe the design method of offline ILC in detail.
Point 5: in the text, you refer to some important assumptions or knowledge without citation, e.g., line 270-271, line 360-361
Response 5: We added these citations to the revised manuscript
Point 6: To the best of my knowledge, the ILC is used to minimize the error to zero in the predefined window. In such a case, I do not understand what the ILC is used to, due to the fact that you obtained overshoot. ILC allows obtaining perfect tracking.
Response 6: We use ILC to obtain the control variable corresponding to the motor cogging torque, and then compensate this control variable offline into the system (refer to Section 3.4 of the article). Therefore, ILC is not used in the final control scheme proposed in this paper.
Point 7: What was the goal of the proposed control? Please improve the definition, of what you want to achieve by using the developed method. You mention about compensation of cogging torque in PMSM drive, But you do not present the comparison to clear PI control structure without compensation. I am asking for this because I do not understand the relation of torque load for checking the robustness.
Response 7: The goal of the control is to minimize the pointing error when the system scans at a uniform speed, which is explained in Line 432~441 of the revised manuscript. We compared ILC and PI in Section 3.3. For off-line compensation, we regard it and NECOD compensation as a composite disturbance suppression strategy, so there is no special comparative experiment. It can be seen from equation 1 that the system model is related to the moment of inertia, and the moment of inertia is related to the size of the load. Therefore, changing the load can change the model of the system. Our robustness experiment is to verify the performance of the controller when the system model changes.
Point 8: There is no additional plot in robustness experiments.
Response 8: Many experimental results are given in sections 5.1 ~ 5.2 of the paper. Because of the length of the paper and the curve is not very necessary for robustness analysis, we only give the final statistical data.
Reviewer 2 Report
This paper presents a method to suppress the cogging torque of a PMSM used in a photoelectric stabilization platform. The title of the paper should be changed to better summarize the work done in this study.
The authors should explain why another type of machine, without cogging torque was not considered for this application? For example, synchronous reluctance machine.
The English spelling should be improved. As example, this subtitle is difficult to understand :"4. Iterative Learning Based Negative Effect Compensation Control of Disturbance"
line 45 - "and the cogging torque changes with the position of the motor" should be corrected with "and the cogging torque changes with the position of the rotor relative to the stator"
line 326 "The specific principle block diagram is shown in the Figure 15." I think is Fig. 11. Other Figure numbers mismatch exist in text.
Overall, the work done is valuable, but the presentation should be improved.
Author Response
We are truly grateful to your suggestions. Based on these comments, we have made careful modifications on the manuscript.
Point 1: The authors should explain why another type of machine, without cogging torque was not considered for this application? For example, synchronous reluctance machine.
Response 1: In Line 38~41 of the revised manuscript, we added the basis for motor selection.
Point : The English spelling should be improved. As example, this subtitle is difficult to understand :"4. Iterative Learning Based Negative Effect Compensation Control of Disturbance"
Response : We revised some spelling in the manuscript. The subtitle of section 4 has been changed to “Design and Analysis of Negative Effect Compensation Control of Disturbance”.
Point : line 45 - "and the cogging torque changes with the position of the motor" should be corrected with "and the cogging torque changes with the position of the rotor relative to the stator"
Response : Corrected in modified manuscript.
Point : line 326 "The specific principle block diagram is shown in the Figure 15." I think is Fig. 11. Other Figure numbers mismatch exist in text.
Response : We rechecked and corrected the serial numbers of all figures and tables in the article.
Reviewer 3 Report
This paper introduces some active anti-interference technology on the high torque output efficiency of PMSM. Then, it suggests a method of combining off-line iterative learning control and negative effect compensation of disturbance for cogging torque problems. And the superior of this paper is through comparing the Los control accuracy with the existing method. The disturbance suppression topic is meaningful in the iterative control field, the reviewer has the following suggestions.
-
Fig.2 shows some surveillance cameras, the author is advised to mark the shafting size directly.
-
In line 89, abbreviations should be listed in their entirety the first time they appear.
-
The author is suggested to list the innovations and contributions of this work in the paper.
-
Most of the selected references should be published no longer than 5 years, the author is suggested to update them.
-
Include references where necessary, such as line-359.
Author Response
We are truly grateful to your suggestions. Based on these comments, we have made careful modifications on the manuscript.
Point 1: Fig.2 shows some surveillance cameras, the author is advised to mark the shafting size directly.
Response 1: We indicated the size range of shafting in the annotation of Figure 2, but for the size of each platform, because most of them are confidential, it is impossible to verify.
Point 2: In line 89, abbreviations should be listed in their entirety the first time they appear.
Response 2: Corrected in revised manuscript.
Point 3: The author is suggested to list the innovations and contributions of this work in the paper.
Response 3: In the conclusions of the paper, we illustrate the innovations and contributions.
Point 4: Most of the selected references should be published no longer than 5 years, the author is suggested to update them.
Response 4: Updated in revised manuscript.
Point 5: Include references where necessary, such as line-359.
Response 5: References have been added where necessary
Round 2
Reviewer 1 Report
It can be accepted.